# Spatiotemporal Variability of Precipitation in Beijing, China during the Wet Seasons

**Meifang Ren** [1,2]**, Zongxue Xu** [1,2,*]**, Bo Pang** [1,2,*]**, Jiangtao Liu** [1,2]  **and Longgang Du** [3]

1   College of Water Sciences, Beijing Normal University, Beijing 100875, China;
    renmeifang@mail.bnu.edu.cn (M.R.); liujiangtao@mail.bnu.edu.cn (J.L.)
2   Beijing Key Laboratory of Urban Hydrological Cycle and Sponge City Technology, Beijing 100875, China
3   Beijing Hydrology Bureau, Beijing 100038, China; dulonggang@139.com
*   Correspondence: zxxu@bnu.edu.cn (Z.X.); pb@bnu.edu.cn (B.P.); Tel.: +86-10-58801989 (B.P.)

**Abstract:** To comprehensively evaluate the changes in precipitation patterns in the context of global climate change and urbanization, the spatiotemporal variability of precipitation during the wet seasons of 1981–2017 in Beijing was analyzed in this study using up-to-date daily and hourly precipitation data from observation stations. It was concluded that the average annual precipitation in wet seasons showed a downward trend, while the simple daily intensity index (SDII) showed an upward trend. Precipitation in the central urban area of Beijing showed obvious changes from 1981 to 2017; the average annual precipitation in the central urban area was almost as great as that in Miyun country after 2010, which was the storm center for the past three decades. The average annual maximum 3-h and 6-h precipitation in the 2010s was higher than the past three decades, especially in urban and suburban areas. In addition, the atmospheric circulation index, urbanization impact, and topography were all found to be important factors that affect the pattern of precipitation in Beijing.

**Keywords:** precipitation; trends; atmospheric index; urbanization; topography; Beijing

## 1. Introduction

Global and regional precipitation patterns have changed in the context of global climate change and urbanization. The frequency of extreme hydrological events, such as flood events and storms, has increased in the past decades [1–3]. Annual global economic losses caused by flood events reached approximately 300 billion dollars in the past decades, and some of the heaviest losses were related to extreme precipitation events in Asia [4–6]. The Fifth Assessment Report of the Intergovernmental Panel on Climate Change identified that global climate change affects global extreme events (e.g., extreme temperature and extreme precipitation), and it predicted that such extreme climate events would continue to occur until the end of the 21st century [7]. Therefore, many studies have been undertaken to quantify the trends and changes of precipitation at the global, regional, and watershed scales. It has been demonstrated that, globally, land precipitation has increased by 2% since the end of the 20th century. However, the occurrence of extreme precipitation events has increased significantly, including in some regions where there has been no change in the total amount of precipitation [8–10]. Groisman et al. [11] highlighted that trends of increase in the occurrence of extreme precipitation events have been found in China, the United States, Canada, Poland, Mexico, Norway, and other countries. Lu et al. [12] pointed out that extreme precipitation in China has increased in association with global climate change, and that the occurrence of extreme precipitation events in the region of the North China Plain has become more random.

Urban areas are sensitive to extreme meteorological events because such areas are centers of population and infrastructure [13]. Therefore, it is necessary to analyze the changes of precipitation patterns and spatiotemporal variations of extreme precipitation in metropolitan areas in the context of a changing environment [14]. As the political, economic, and cultural center of China, Beijing has undergone very rapid urbanization in the past decades, and it has been demonstrated that urban development has led to an increase in the intensity of precipitation, especially in downwind urban regions [15,16]. Two principal conclusions have been drawn following previous studies of the variations of precipitation in Beijing. The first is that both annual precipitation and extreme precipitation in Beijing showed a downward trend in the past decades. For example, Song et al. [17] found that the frequency, amount, and contributions of extreme precipitation events in Beijing have had significant downward trends in the previous 50 years (1960–2012). The second conclusion is that the phenomenon of a "precipitation island effect" has occurred in Beijing. For example, Zhai et al. [18] and Zhen et al. [19] suggested that the average annual precipitation in the central urban area in Beijing is greater than that in the surrounding areas because of urbanization effects.

Apart from urbanization, the atmospheric circulation index and local topography have also been recognized as important factors in the distribution of precipitation [17]. The objective of this study is to provide a better understanding of the characteristics of precipitation patterns and variations of extreme precipitation in Beijing (in particular, Beijing suffered heavy rainstorms in 2012 and 2016), and examine the influence of the atmospheric circulation index and local factors. The spatiotemporal variations of precipitation during 1981–2017 in Beijing were analyzed in this study, using up to date daily and hourly precipitation data from observation stations and multiple analysis methods. The precipitation indices recommended by the Expert Team on Climate Change Detection and Indices (ETCCDI) of the World Meteorological Organization were used in this study, as these have been widely used to evaluate the impact of climate change at the global scale [20] and in many parts of the world [21–25]. By comparing the differences in precipitation patterns between different subareas, the existence of a "precipitation island effect" in Beijing was further investigated and identified.

## 2. Data and Methodology

### 2.1. Study Area Description

Beijing is located in northern China (Figure 1). Beijing has a typical warm temperature semi-humid continental monsoon climate, which has hot rainy summers, and cold dry winters. In Beijing, the average annual temperature is 11.7 °C; the highest temperature in summer can reach 42.6 °C and the lowest temperature in winter can reach −27.4 °C [17,19,26]. Precipitation is distributed unevenly across the different seasons. Precipitation in summer (June–August) accounts for more than 70% of the annual precipitation, spring and autumn account for approximately 28%, and winter is largely dry [27]. The average annual precipitation in Beijing ranges from 372.1 to 682.9 mm/year at different observed stations, based on daily observed precipitation data at 30 stations from 1981 to 2017. The spatial distribution of precipitation is highly heterogeneous. For example, Miyun and Pinggu counties, which are located in the northeast of Beijing, receive the most precipitation, followed by central urban areas and near southern suburban areas. The western mountainous area and southeastern suburban area receive the least precipitation. The spatial distribution of average annual precipitation in Beijing is shown in Figure 1.

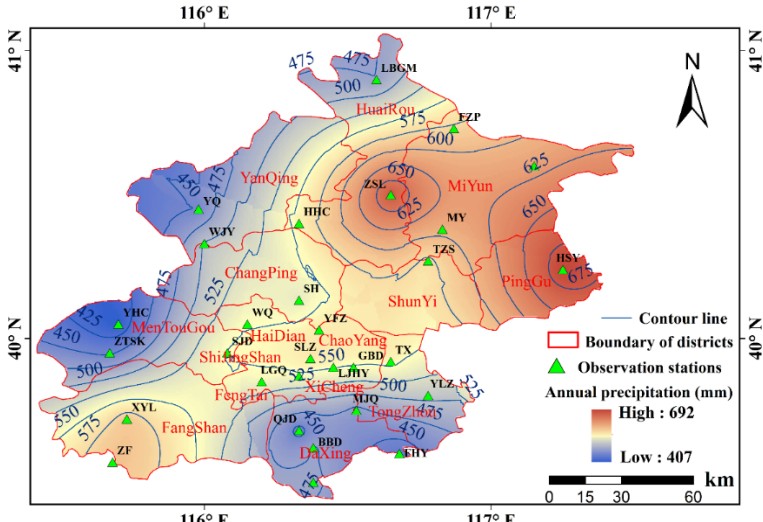

**Figure 1.** Location of observation stations and spatial distribution of average annual precipitation in Beijing.

*2.2. Data Description*

2.2.1. Precipitation Data

In this study, daily precipitation data from 30 observation stations (1981–2017) and hourly precipitation data from eight observation stations were used for analysis and calculation. All data were obtained from the Beijing Hydrology Bureau. Strict quality control was performed on both daily and hourly precipitation data. The number of missing data at all stations was less than 0.1% of the total data. The average daily precipitation at near stations was used as a substitute for the missing data. The accuracy of both datasets was 0.01 mm.

In this study, to reflect the characteristics of precipitation in different subareas characterized by distinct topographic conditions, Beijing was divided into six different subareas [17,28]: the central urban area, near northern suburb, near southern suburb, far suburb (northeastern), southwest mountainous area, and northwest mountainous area. Information regarding the 30 observation stations is presented in Table 1.

**Table 1.** Information regarding the observation stations.

| Regions | Name of Stations | Elevation (m) | Regions | Name of Stations | Elevation (m) |
|---|---|---|---|---|---|
| Central urban area | Gaobeidian (GBD) | 36 | Near southern suburb | Banbidian (BBD) | 33 |
| | Lejiahuayuan (LJHY) | 44 | | Fengheying (FHY) * | 20 |
| | Lugouqiao (LGQ) * | 64 | | Majuqiao (MJQ) | 30 |
| | Songlinzha (SLZ) * | 55 | | Nangezhuang (NGZ) | 32 |
| | Wenquan (WQ) * | 54 | | Yulinzhuang (YLZ) | 21 |
| | Tongxian (TX) | 28 | Far suburb | Fanzipai (FZP) | 520 |
| | Yangfangzha (YFZ) | 44 | | Huangsongyu (HSY) | 173 |
| | You'an men (YAM) | 46 | | Miyun (MY) * | 74 |
| Near northern suburb | Shahe (SH) | 39 | | Tangzhishan (TZS) | 46 |
| Southwest mountainous area | Sanjiadian (SJD) | 175 | Northwest mountainous area | Xiahui (XH) | 233 |
| | Wangjiayuan (WJY) * | 761 | | Huanghuacheng (HHC) * | 271 |
| | Xiayunling (XYL) | 446 | | Labagoumen (LBGM) | 492 |
| | Yanhecheng (YHC) | 539 | | Qianjiadian (QJD) | 40 |
| | Zhaitangshuiku (ZTSK) | 560 | | Yanqing (YQ) | 495 |
| | Zhangfang (ZF) | 109 | | Zaoshulin (ZSL) | 325 |

* Indicates the station records for daily and hourly precipitation data.

### 2.2.2. Land Use Data

Land use maps of Beijing in 1980, 1990, 2000, 2005, 2014, and 2017, based on remote sensing imagery, were used in this study. These data were obtained from both the remote sensing monitoring database of land use status in China (Data Center for Resources and Environmental Sciences of the Chinese Academy of Sciences), and the Landsat remote sensing image database (www.gscloud.cn). The image data from 1980 were collected from the remote sensing monitoring database of land use status in China, which is based on interpretations of Moderate Resolution Imaging Spectroradiometer (MODIS) and Landsat Thematic Mapper (Landsat-TM) satellite remote sensing images [29]. The resolution of this image data is 1 km. The data of the remaining years were derived from Landsat series images, and the resolution of these images data are 30 m.

### 2.3. Methodology Description

#### 2.3.1. Extreme Precipitation Indices

The Expert Team on Climate Change Detection and Indices of the World Meteorological Organization recommended 27 core indices to represent extreme temperatures and precipitation. In this study, annual mean precipitation and nine extreme precipitation indices are adopted to analyze the spatiotemporal variation of extreme precipitation in Beijing. The name and meaning of each extreme precipitation indices are presented in Table 2. These indices are used widely in research related to extreme precipitation change [20,30,31].

**Table 2.** Indices of extreme precipitation.

| Code | Descriptive Name | Definition of the Indices | Units |
|------|------------------|---------------------------|-------|
| AMP | Annual mean precipitation | Annual mean precipitation | mm |
| SDII | Simply daily intensity index | Annual mean precipitation/total number of wet days | mm/day |
| R20mm | Number of moderate precipitation days | Annual precipitation days with daily precipitation greater than 20 mm | days |
| R50mm | Number of violent precipitation days | Annual precipitation days with daily precipitation greater than 50 mm | days |
| Rx1day | Maximum 1-day precipitation amount | Annual maximum 1-day precipitation | mm |
| Rx5day | Maximum 5-day precipitation amount | Annual maximum five consecutive days of precipitation | mm |
| R95p | Precipitation on very wet days | Annual precipitation exceeds 95% threshold | mm |
| R99p | Precipitation on extremely wet days | Annual precipitation exceeds 99% threshold | mm |
| CWD | Maximum consecutive wet days | Maximum number of consecutive days with daily precipitation greater than or equal to 1.0 mm | days |
| CDD | Maximum consecutive dry days | Maximum number of consecutive days with daily precipitation less than 1.0 mm | days |

The percentage threshold method is widely used to define extreme precipitation events; thus the 95% and 99% thresholds were used to define extreme precipitation events in this study. Specifically, the daily precipitation of wet days was arranged in ascending order. Then, the 95% and 99% precipitation values were defined as the threshold values of extreme precipitation at each station. It was considered that an extreme precipitation event had occurred when the daily precipitation at a station reached or exceeded these thresholds on a certain day [17]. In this study, a value of 0.1 mm/d was used as the threshold to define the occurrence of a precipitation event.

### 2.3.2. Trend Analysis Methods

This study adopts several methods to analyze the spatiotemporal variation of extreme precipitation in Beijing: the linear fitting method, 5-year moving average method, spatial interpolation, and Mann–Kendall (M-K) nonparametric test method. The Kriging interpolation method was used to calculate the spatial distribution of annual precipitation in this study. The M-K test is a nonparametric test method used widely in hydrological and meteorological research. It is commonly used to test the trend of a dataset that does not obey any specific distribution. The calculation formula of the M-K method is as follows:

$$Z = \begin{cases} \frac{S-1}{\sqrt{\text{var}(S)}}, & S > 0 \\ 0, & S = 0 \\ \frac{S+1}{\sqrt{\text{var}(S)}}, & S < 0 \end{cases} \tag{1}$$

$$S = \sum_{i=1}^{n-1} \sum_{k=i+1}^{n} \text{sgn}(x_k - x_i) \tag{2}$$

$$\text{sgn}(x_k - x_i) = \begin{cases} 1, x_k - x_i > 0 \\ 0, x_k - x_i = 0 \\ -1, x_k - x_i < 0 \end{cases} \tag{3}$$

where $n$ is the length of the dataset, $x_k$ and $x_i$ are the sequences of the sample data, and *var(S)* is the variance of the statistical variable $S$. A positive (negative) value of $Z$ represents an upward (a downward) trend of the detected dataset.

### 2.3.3. Correlation Analysis

The Spearman's correlation coefficient method was used to calculate the correlation coefficient between the extreme precipitation indices and the atmospheric circulation index. The larger the absolute value of the correlation coefficient (closer to 1), the stronger the correlation between indices.

## 3. Results Analysis

### 3.1. Annual Precipitation Description

The temporal variations of the average annual precipitation in the entire year and in wet seasons (June–September) at the 30 observation stations in Beijing (1981–2017) are shown in Figure 2. It can be seen that the changes in the trend of average annual precipitation in the entire year were consistent with those in wet seasons during the study periods; that is, both showed downward trends during 1981–2017. Figure 3 shows the average annual precipitation in the entire year and that in wet seasons per decade at the 30 observation stations. It can be seen that precipitation in Beijing in the 2000s showed a downward trend compared with the 1980s and 1990s, whereas it showed an obvious upward trend during 2010–2017.

Precipitation events in the wet seasons (June–September) are the focus of this study because extreme precipitation in Beijing mainly occurs during the wet seasons. Figure 4 shows the spatial distribution of annual precipitation in the wet seasons in different decades, interpolated by the original Kriging interpolation method. It is notable that precipitation in the central area showed obvious changes from the 1980s to the 2010s. The maximum annual precipitation in the 1980s and 1990s appeared in Miyun county, which is located in the northeast of Beijing. However, in the 2000s, the storm center gradually extended to the central urban area, and at the beginning of the 21st century the central area became the second storm center in Beijing, since its annual precipitation was almost as great as that in Miyun county.

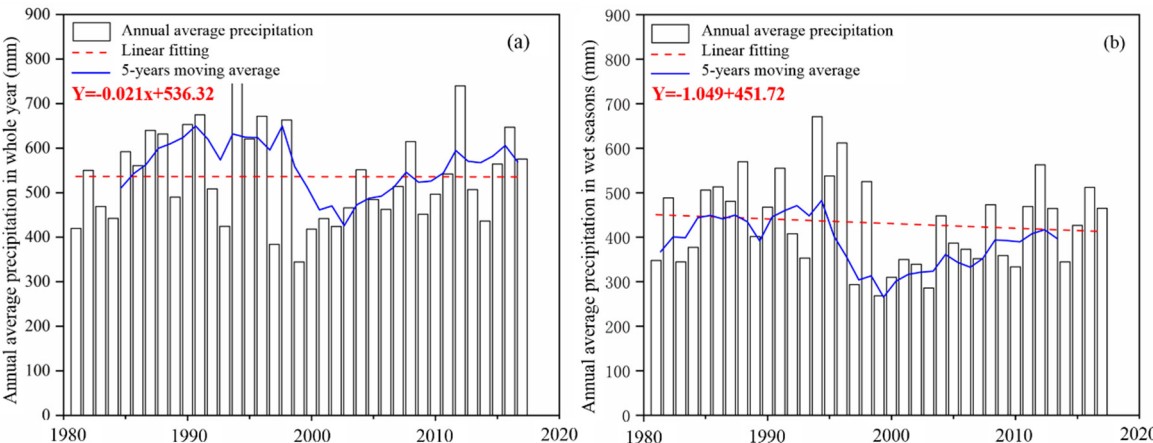

**Figure 2.** Temporal variation of average annual precipitation across the whole year (**a**) and in the wet seasons (**b**).

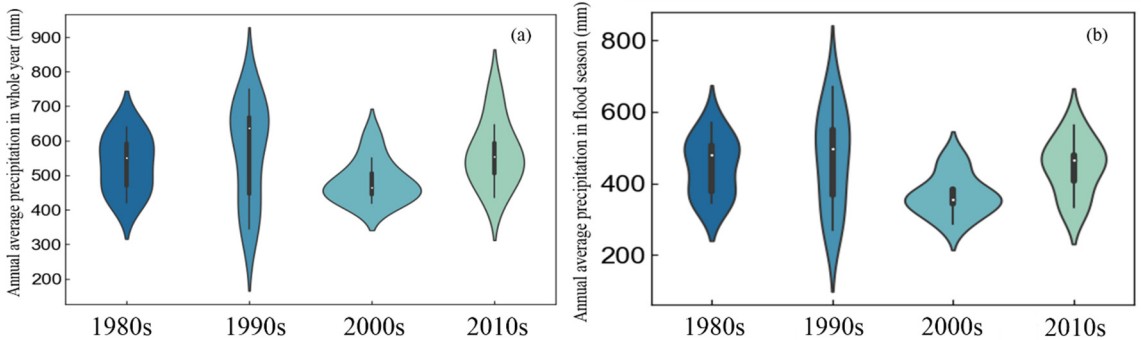

**Figure 3.** Trends of average annual precipitation per decade across the whole year (**a**) and in the wet seasons (**b**).

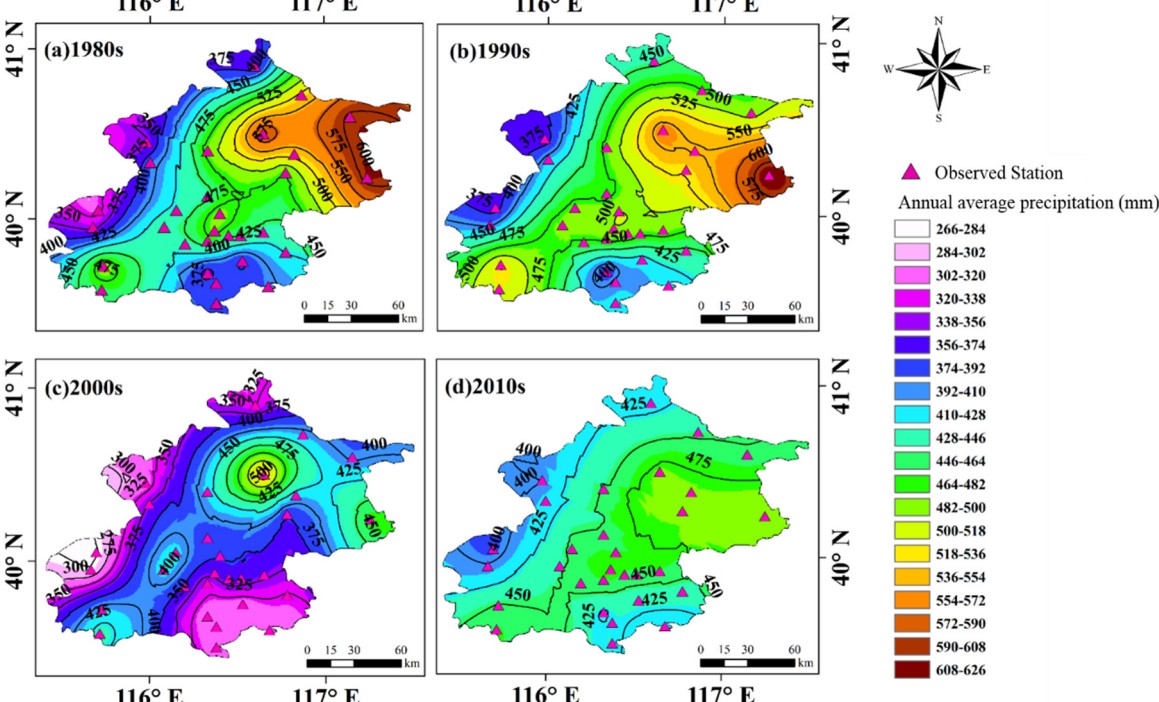

**Figure 4.** Spatial distribution of annual precipitation in Beijing (wet seasons) for (**a**) 1980s, (**b**) 1990s, (**c**) 2000s and (**d**) 2010s.

### 3.2. Temporal Trends of Daily Precipitation Indices in the Wet Seasons

The temporal changes of the average precipitation indices at the 30 observation stations in Beijing during 1981–2017 are shown in Figure 5. The annual mean precipitation (AMP), simple daily intensity index (SDII), number of moderate precipitation days (R20mm), and number of violent precipitation days (R50mm) indices reflect the overall condition of precipitation, while the others mainly focus on extreme precipitation. Four indices (the AMP, maximum 5-day precipitation amount (Rx5day), R20mm, and maximum consecutive wet days (CWD) indices) showed downward trends, while the SDII, maximum 1-day precipitation amount (Rx1day), R50mm, precipitation on very wet days (R95p), precipitation on extremely wet days (R99p), and maximum consecutive dry days (CDD) indices showed an upward trend.

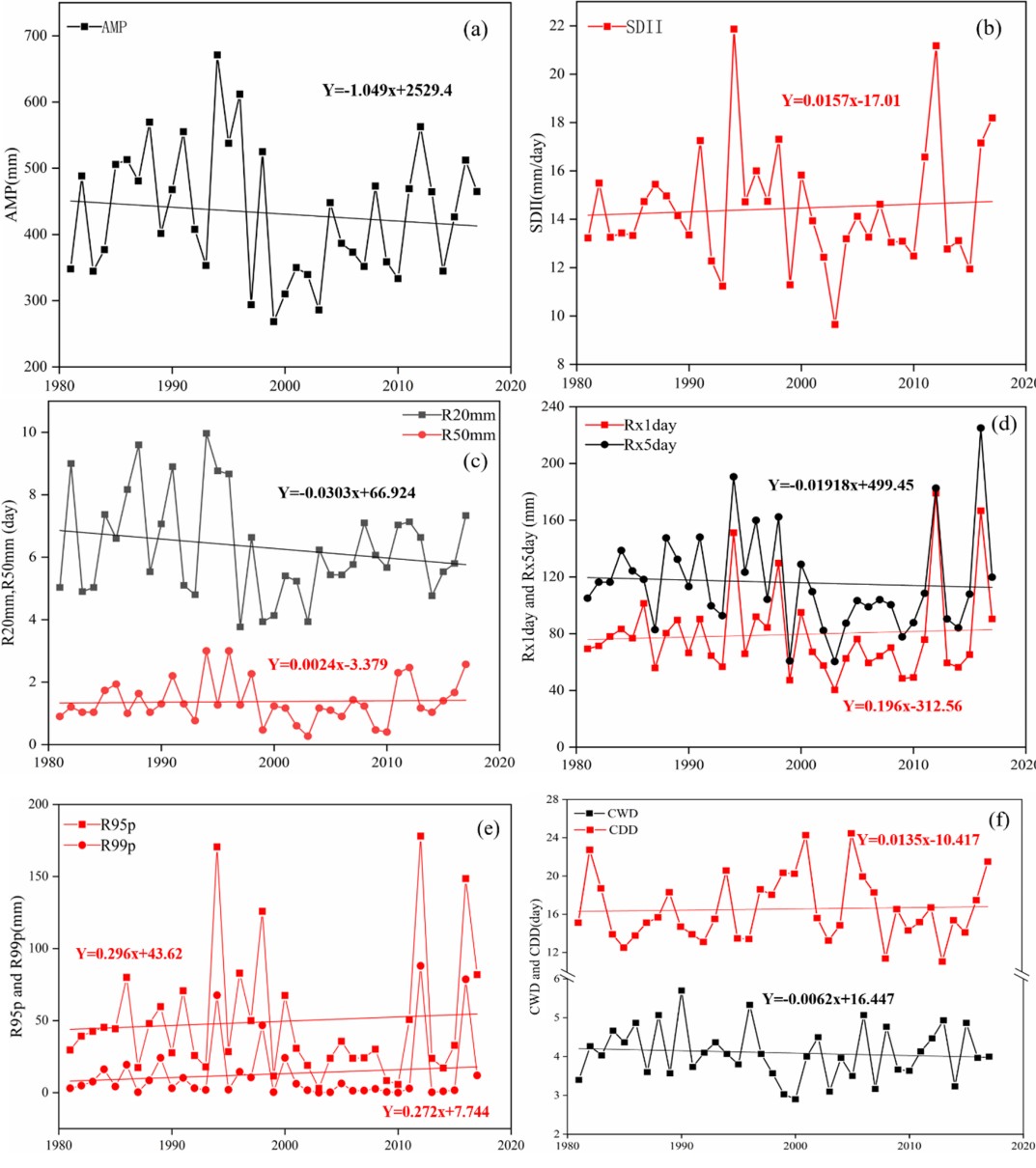

**Figure 5.** Temporal trends of the precipitation indices during 1981–2017 for (**a**) annual mean precipitation (AMP), (**b**) simple daily intensity index (SDII), (**c**) number of moderate and violent precipitation days (R20mm and R50mm), (**d**) maximum 1-day and 5-day precipitation amount (Rx1day and Rx5day), (**e**) precipitation on very and extremely wet days (R95p and R99p), (**f**) maximum consecutive dry days (CWD) and maximum consecutive wet days (CDD).

To reflect the precipitation patterns in different parts of Beijing, the average precipitation indices in different subareas during the wet seasons were estimated (Table 3). The average AMP indices value for all stations was 430.53 mm; however, this value varied from 385.7 to 493.5 mm in different subareas, with the highest value in the far suburb, followed by the central urban area. The average SDII for all stations was 14.40 mm/d (range: 12.96–15.66 mm/d). The far suburb had the highest value of the SDII (15.66 mm/d), followed by the central urban area (15.31 mm/d). The SDII values in the near northern and southern suburb (14.38 and 14.85 mm/d, respectively) were higher than in the southwestern and northwestern mountainous areas (13.21 and 12.96 mm/d, respectively). The values of the R20mm and R50mm indices varied in the range of 5.49–7.55 and 1.13–1.74 d, respectively, and similar to the AMP and the SDII, the highest values of both the R20mm and R50mm indices were in the far suburb, followed by the central urban area.

**Table 3.** Average precipitation indices in different areas in the wet seasons.

| Areas | Precipitation Indices | | | | | | | | | |
|---|---|---|---|---|---|---|---|---|---|---|
| | AMP mm | SDII mm/day | R20mm day | R50mm day | Rx1day mm | Rx5day mm | R95p mm | R99p mm | CWD day | CDD day |
| Urban area | 447.43 | 15.31 | 6.80 | 1.52 | 85.70 | 121.76 | 46.93 | 15.86 | 3.98 | 17.19 |
| North suburb | 429.36 | 14.38 | 6.19 | 1.22 | 80.54 | 115.38 | 46.32 | 10.52 | 4.11 | 17.35 |
| South suburb | 385.70 | 14.85 | 5.81 | 1.28 | 77.04 | 110.95 | 35.98 | 11.23 | 3.72 | 18.30 |
| Far suburb | 493.05 | 15.66 | 7.55 | 1.74 | 86.10 | 128.79 | 47.50 | 11.71 | 4.12 | 15.21 |
| Southwest mountainous area | 412.63 | 13.21 | 5.49 | 1.19 | 77.56 | 113.35 | 48.26 | 13.67 | 4.36 | 16.04 |
| Northwest mountainous area | 415.03 | 12.96 | 5.79 | 1.13 | 66.85 | 102.86 | 35.34 | 10.73 | 4.30 | 15.48 |
| All stations | 430.53 | 14.40 | 6.27 | 1.35 | 78.96 | 115.51 | 43.39 | 12.29 | 4.10 | 16.59 |

The Rx1day and Rx5day indices varied in the range of 66.85–86.10 and 102.86–128.79 mm/d, respectively, with the highest values in the far suburb, followed by the central urban area. The average values of the R95p and R99p indices were in the range of 35.34–48.26 and 10.52–15.86 mm, respectively. Notably, the central urban area had the highest value for the R99p indices. The maximum values of the CWD indices were in the mountainous areas and the far suburb area, while the lowest values were in the central urban area and near southern suburb. The highest values of CDD were in the suburb and central urban area.

In general, the highest values of the precipitation indices that reflect the overall condition of precipitation (the AMP, SDII, R20mm, and R50mm) were found in the far suburb, followed by the central urban area. The value of the R99p indices in the central urban area was obviously higher than that in other areas. In addition, the central urban area had a relatively lower value for the CWD indices and higher value for the CDD indices.

*3.3. Spatial Patterns of Precipitation Indices in the Wet Seasons*

Table 4 lists the average values of the nonparametric test statistic index Z in different areas of Beijing, and Figure 6 presents the spatial distribution of the Z value of the precipitation indices at each station. The average Z value of the AMP indices for all stations was −0.56, and it ranged from −1.35 to 0.07 in different subareas. The AMP indices in the urban area showed a slight upward trend (the Z value was positive), while it showed a downward trend in other subareas. It can be seen from Figure 6 that 40% of the stations (12 stations) presented an upward trend of the AMP indices (most of these stations were located in the central urban area and the southern suburb). The stations in the urban area (eight stations), except the Lugouqiao (LGQ) and You'an men (YAM) stations, showed an upward trend, and among the five stations in the southern suburb, the Majuqiao (MJQ), Banbidian (BBD), and Nangezhuang (NGZ) stations also showed an upward trend. The average Z value of the SDII for all stations was 0.06, and it ranged from −0.29 to 0.17 in different subareas. In all subareas except

the far suburb, the SDII presented a slightly upward trend; that is, approximately 53% (16 stations) of stations presented a positive Z value. The average Z value of the R20mm and R50mm indices varied from −1.58 to −0.47 and from −0.87 to 0.36, respectively, in different subareas. Notably, the R50mm indices at 50% of stations showed an upward trend, and these stations were located in the central urban area, southern suburb, and mountainous areas, while all stations presented a downward trend of the R20mm indices.

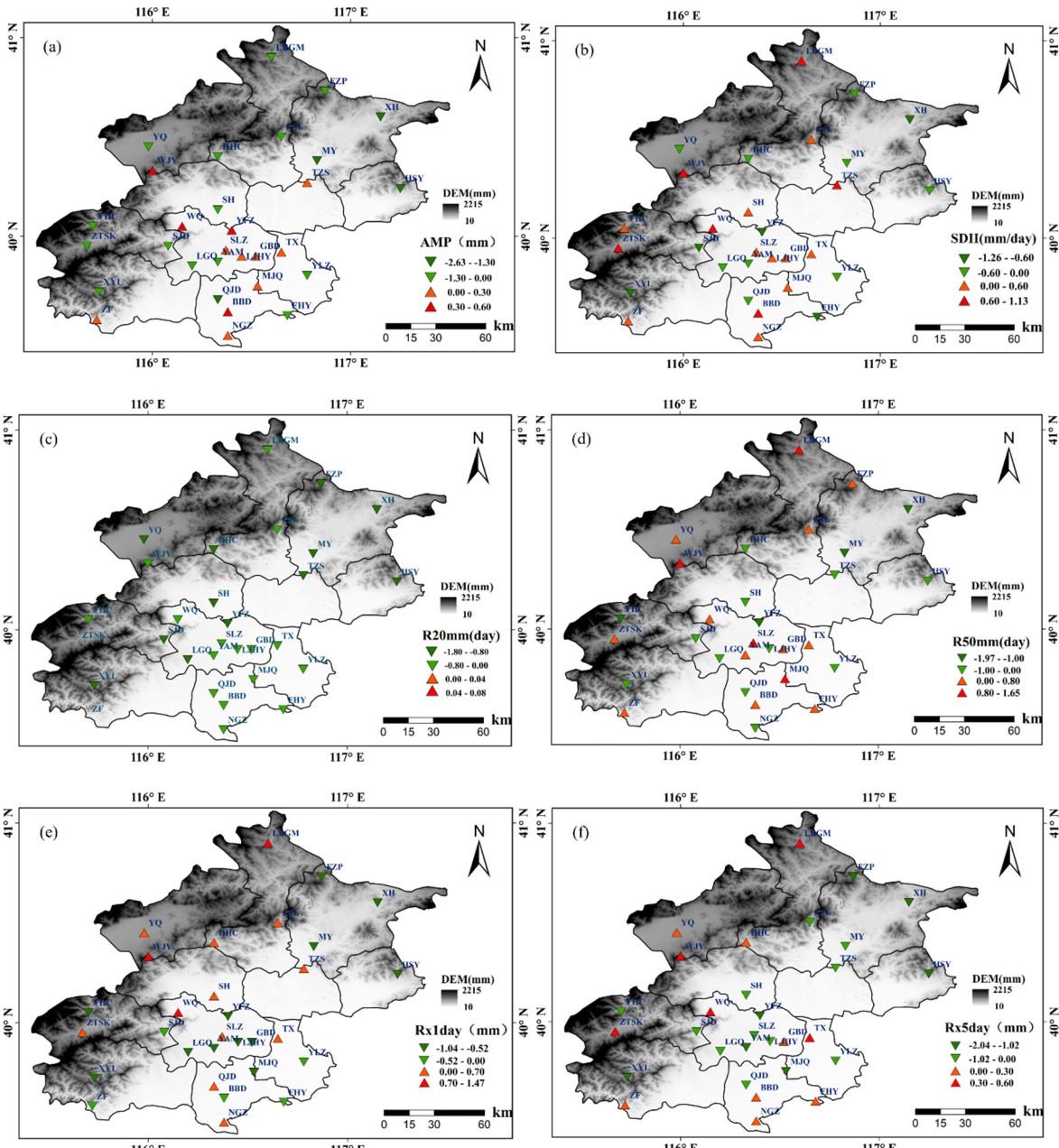

**Figure 6.** *Cont.*

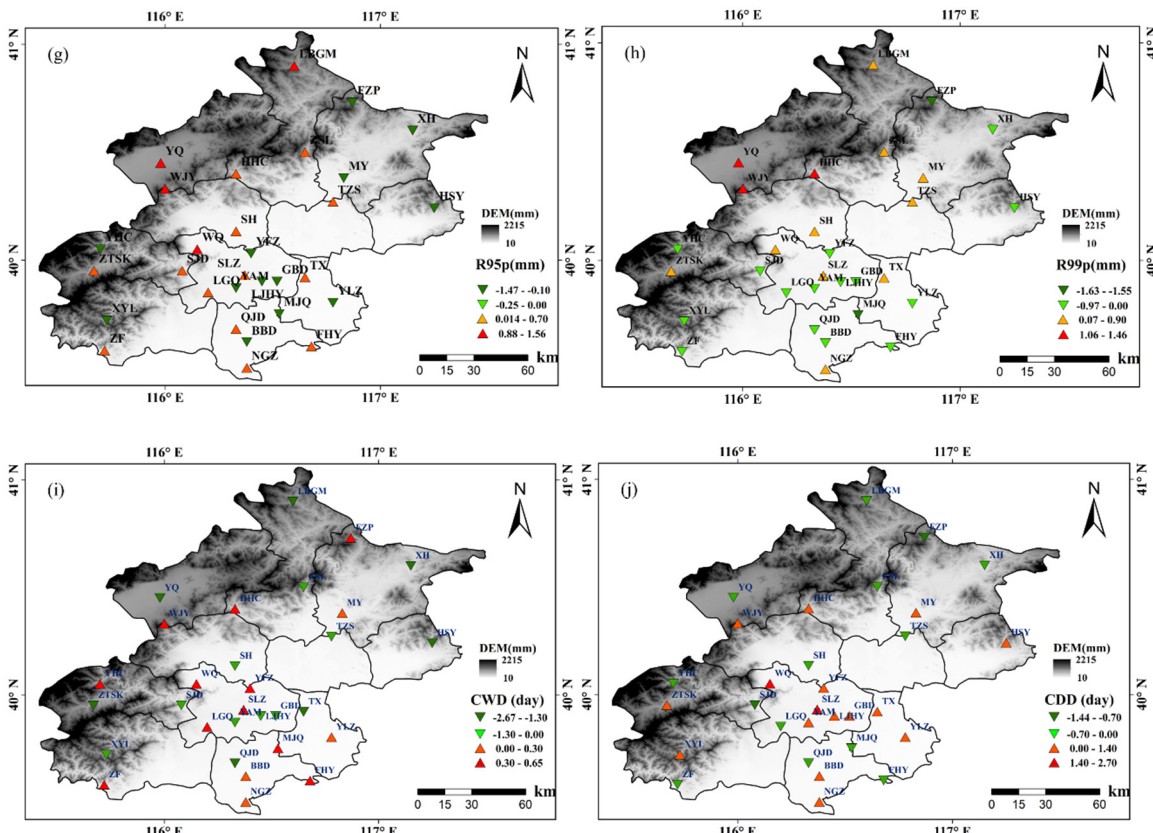

**Figure 6.** Spatial distribution of the Z value of extreme precipitation indices for (**a**) annual mean precipitation (AMP), (**b**) simple daily intensity index, (**c**) number of moderate precipitation days (R20mm), (**d**) number of violent precipitation days (R50mm), (**e**) maximum 1-day precipitation amount (Rx1day), (**f**) maximum 5-day precipitation amount (Rx5day), (**g**) precipitation on very wet days (R95p), (**h**) precipitation on extremely wet days (R99p), (**i**) maximum consecutive dry days (CWD) and (**j**) maximum consecutive wet days (CDD).

**Table 4.** Trends of the precipitation indices in different subareas (Z).

| Areas | Precipitation Indices | | | | | | | | | |
|---|---|---|---|---|---|---|---|---|---|---|
| | AMP mm | SDII mm/day | R20mm day | R50mm day | Rx1day mm | Rx5day mm | R95p mm | R99p mm | CWD day | CDD day |
| Urban area | 0.07 | 0.04 | −0.53 | 0.02 | −0.29 | −0.52 | −0.17 | −0.12 | −0.24 | 1.13 |
| North suburb | −0.77 | 0.07 | −1.58 | −0.09 | 0.42 | −0.72 | 0.22 | 0.65 | −0.84 | −0.66 |
| South suburb | −0.13 | 0.17 | −0.47 | 0.28 | −0.30 | −0.30 | −0.20 | −0.49 | 0.27 | −0.10 |
| Far suburb | −1.35 | −0.29 | −1.22 | −0.87 | −0.56 | −1.16 | −0.60 | −0.41 | −0.81 | −0.12 |
| Southwest mountainous area | −0.28 | 0.16 | −0.63 | 0.10 | −0.10 | −0.18 | −0.16 | 0 | −0.32 | −0.04 |
| Northwest mountainous area | −0.89 | 0.21 | −0.87 | 0.36 | 0.66 | −0.09 | 0.83 | 0.70 | −1.34 | −0.17 |
| All stations | −0.56 | 0.06 | −0.88 | −0.03 | −0.03 | −0.50 | −0.08 | 0.33 | −0.55 | 0.01 |

The average Z value of the Rx1day and Rx5day indices varied from −0.56 to 0.66 and from −1.16 to −0.09, respectively, in the six subareas. Overall, 13 (12) stations presented an upward trend of the Rx1day (Rx5day) indices. Most of these stations were located in the western mountainous areas. Most of the stations located in the central urban area and southern suburb showed a downward trend. The average Z value of the R95p and R99p indices was −0.08 and 0.33, respectively. Overall, 57% (17 stations) and 43% (13 stations) of stations presented an upward trend for the R95p and R99p indices, respectively. The average Z value of the CWD indices in the six subareas ranged from −1.34 to

0.27, and this value for all stations was −0.55, which means that the number of continuous maximum wet days in most areas of Beijing was declining. The average Z value of the CDD indices in the six subareas ranged from −0.66 to 1.13 (the average Z value for all stations was 0.01), and the values of the CDD indices in the central urban area showed an obvious upward trend in comparison with the other subareas.

### 3.4. Temporal Trends of Hourly Precipitation

The temporal changing trends of the annual maximum 1-h/3-h/6-h precipitation in different subareas are shown in Figure 7, in which, a, b, c, and d represent the trends in urban area, near suburb area, the far suburb area, and mountainous areas, respectively. It can been seen that the average value of the annual maximum 1-h precipitation in the 2010s in the near suburbs was slightly higher than that in the 2000s, and there were no obvious trends in other subareas; however, the average values of the annual maximum 3-h and 6-h precipitation in the 2010s were higher than those in the past three decades, especially in the urban and suburb areas compared to the mountainous areas; extreme precipitation in 2012 and 2016 could be the main cause for this.

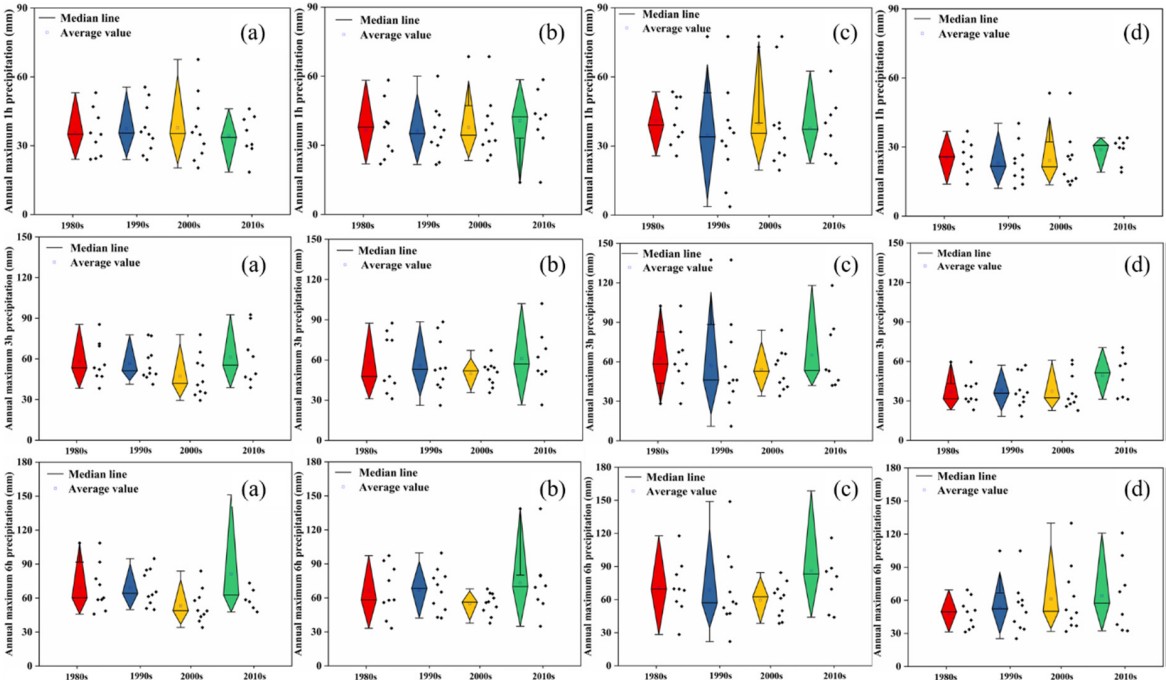

**Figure 7.** The temporal changes of hourly precipitation from the 1980s to 2010s in different subareas for (**a**) urban area, (**b**) near suburb area, (**c**) far suburb area, and (**d**) mountainous areas.

## 4. Discussion

### 4.1. Correlation between the Atmospheric Circulation Index and Precipitation Indices

El Niño and La Niña refer to the phenomenon of abnormal warming and cooling, respectively, of the sea surface in the eastern Pacific Ocean. The Southern Oscillation refers to the phenomenon of reversed abnormal changes of sea level pressure in the Pacific and Indian oceans. EI Niño–Southern Oscillation (ENSO) events are a coupled oceanic–atmospheric phenomenon [32]. Previous studies have shown that ENSO events have considerable impact on extreme precipitation in China, and the East Asian Summer Monsoon (EASM) also has an important moderating effect on precipitation in the eastern parts of China [33,34]. For example, Fu et al. [35] suggested that the annual precipitation and the frequencies of extreme precipitation are correlated with the ENSO index across China, while the relationship is different at different time scales and during different time periods. Miao et al. [36]

concluded that the regional means of R95p were positively correlated with the ENSO across China during 1957–2014. To investigate the influence of atmospheric circulation on precipitation in Beijing, the Spearman's correlation coefficient method was used to analyze the correlation between the extreme precipitation indices and two atmospheric circulation indices—the ENSO index and the EASM index. In this study, because of space limitations, only the correlations between the AMP, SDII, R95p, and R99p precipitation indices and the atmospheric circulation index were estimated. The spatial distribution of the correlation coefficients at each station is shown in Figure 8.

The ENSO index was positively correlated with AMP at all stations, and approximately 87% of stations had positive correlations between the ENSO index and SDII indices. The ENSO index exhibited statistically significant positive correlations with the R95p and R99p indices at 28 and 26 stations, respectively. Conversely, the AMP and SDII indices at most of the stations were negatively correlated with the EASM index, and there were 29 and 28 stations that showed statistically negative correlations with the EASM index. The results indicate that ENSO had a considerable influence on the AMP, SDII, R95p, and R99p indices, while the EASM index had a weakening effect on the AMP, SDII, R95p, and R99p indices in the wet seasons during 1971–2017 in Beijing. These findings are similar to previous research results in Beijing; for example, Wei et al. [37] analyzed the correlation between the extreme precipitation indices and large-scale climate variables in the Beijing–Tianjin sand source region. They found the ENSO and EASM index had a significant impact on precipitation extremes during 1960–2014. Song et al. [14] concluded that there was a significant relationship between the EASM and the extreme precipitation indices in Beijing during the end of the 1970s and in the 1980s.

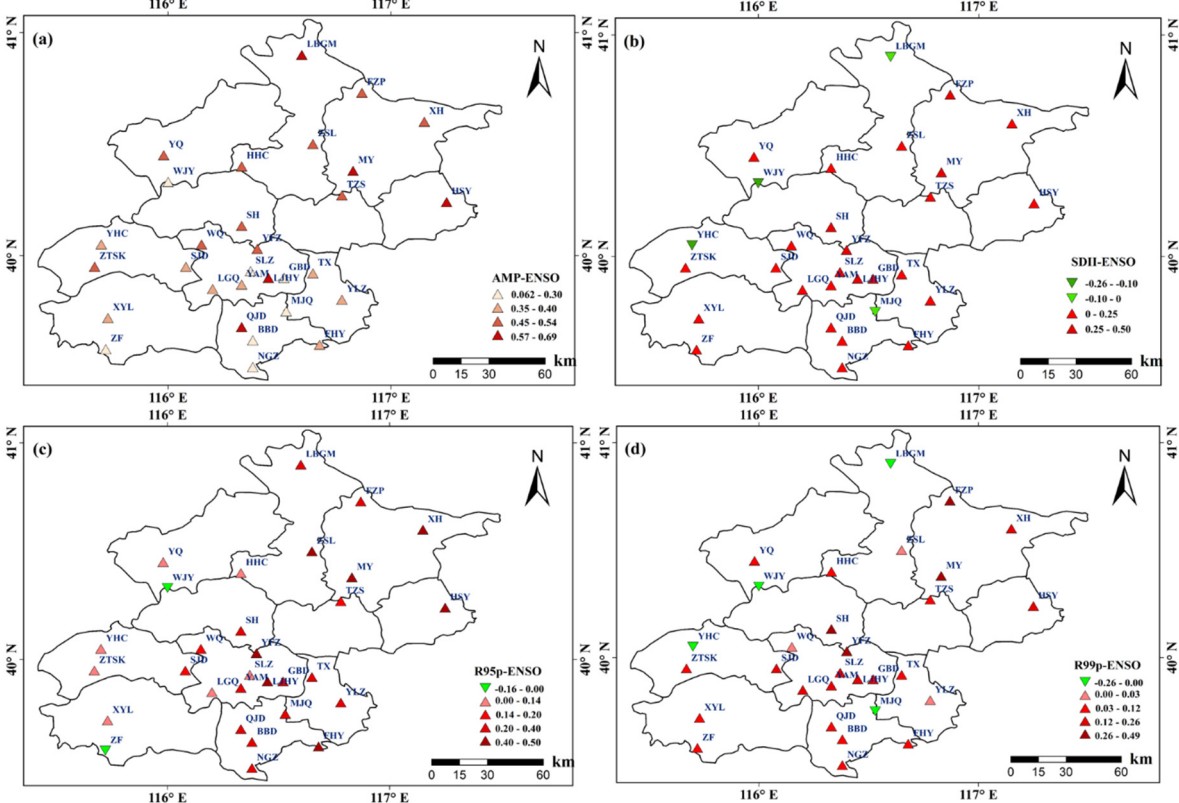

**Figure 8.** *Cont.*

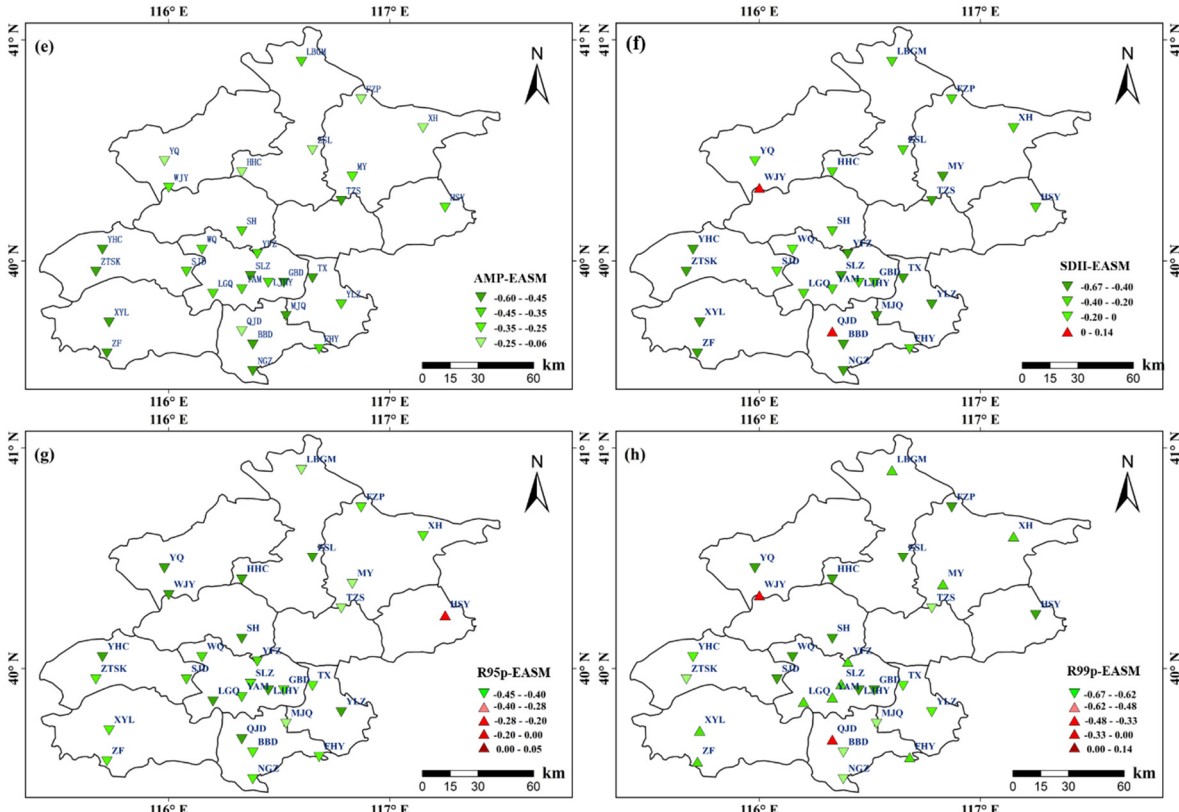

**Figure 8.** Correlation coefficients between precipitation indices and the atmospheric circulation index for (**a**) annual mean precipitation (AMP) and ENSO index, (**b**) simple daily intensity index (SDII) and ENSO index, (**c**) precipitation on very wet days (R95p) and ENSO index, (**d**) precipitation on extremely wet days (R99p) and ENSO index, (**e**) annual mean precipitation (AMP) and EASM index, (**f**) simple daily intensity index (SDII) and EASM index, (**g**) precipitation on very wet days (R95p) and EASM index, and (**h**) precipitation on extremely wet days (R99p) and EASM index.

## 4.2. Impact of Urbanization

The expansion of urbanization is considered a factor that could affect the pattern of wet seasons precipitation in urban and downwind areas. Previous researchers have suggested that the pattern of precipitation in Beijing might reflect the urbanization process and topography (i.e., the western mountainous areas and eastern plain areas). In this study, the impact of urbanization on precipitation was investigated through interpretation of Landsat remote sensing image data. The land use of the Beijing area was divided into four types: water areas, forest areas, farmland areas, and impervious areas. The temporal and spatial changes of land use during 1980–2017 in Beijing are shown in Figure 9a. Forest areas are distributed mainly in the northwest and southeast mountainous areas, urban areas (impervious areas) and farmland areas are distributed in the southeast plain areas, and the amount of water area is very small. The process of urbanization in Beijing has manifested by filling the central areas first, then expanding into surrounding areas. There was no significant increase of impervious areas during 1980–1990. However, the extent of impervious areas expanded rapidly during 1990–2000, at an annual growth rate of 0.48%, and expanded even more quickly (annual growth rate of 0.54%) during 2000–2014, while there was decline during 2014–2017. The impervious areas in 2017 were only 73% of that in 2014, which may reflect the effects of the "Sponge City" construction that begun in Beijing from 2014.

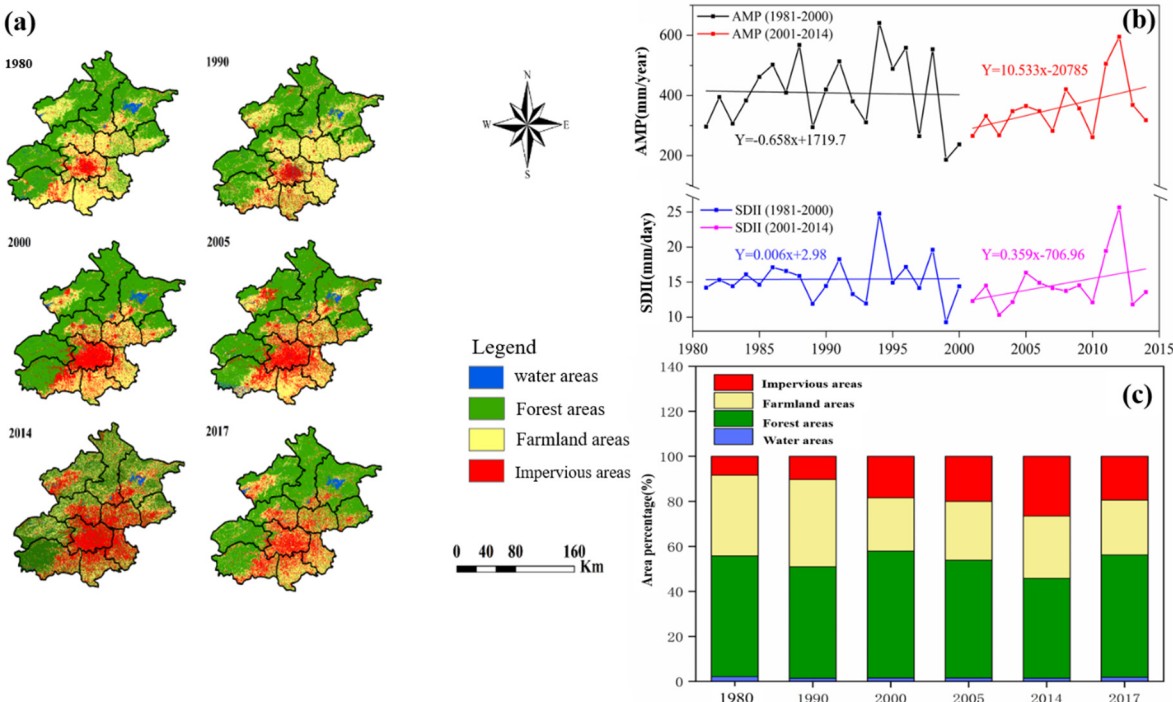

**Figure 9.** Comparison of changes of land use in Beijing (**a**,**c**) and precipitation indices in urban areas (**b**).

With rapid urbanization, the land use pattern of Beijing has changed greatly in the past three decades, especially in terms of the rapid expansion of impervious areas and large-scale reduction of farmland areas. The changes in precipitation amount and intensity in the central urban areas were compared with the variation of impervious areas in Beijing in different decades. As shown in Figure 9, the impervious areas increased by 0.45%/year during 1981–2000 and expanded even more quickly (annual growth rate of 0.54%) during 2000–2014. The AMP indices and the SDII presented obvious upward trends during 2000–2014, and the rate of increase of impervious areas during this period was much faster than in the previous period (1981–2000). It seems that urbanization may have had some impact on the pattern of precipitation in Beijing.

In general, in the previous three decades, the variation in the pattern of precipitation in the central urban area and surrounding areas in Beijing might be related to the change in the impervious areas in Beijing.

### 4.3. Relationship between Elevation and the Precipitation Indices

The relationship between the average precipitation indices and the elevation of each observation station (1981–2017) was analyzed in this study (Figure 10). The red solid line in Figure 10 is the linear fitting line, and "r" is the Spearman's correlation coefficient between the average precipitation indices and elevation. It can be seen that the CWD index was correlated positively with elevation, while the other indices were correlated negatively with the elevation at each station. This means that most precipitation indices (except the CWD index) decreased as the elevation increased. The SDII, CWD, and CDD indices exhibited a strong correlation with elevation; these indices passed the significance test at the 99% confidence level. The Rx1day and R20mm indices passed the significance test at the 95% confidence level. The AMP, Rx5day, R50mm, R95p, and R99p indices had a slightly negative correlation with elevation. These findings are consistent with previous studies that have indicated that topography is one of the most important factors affecting the patterns of precipitation in Beijing [13,16].

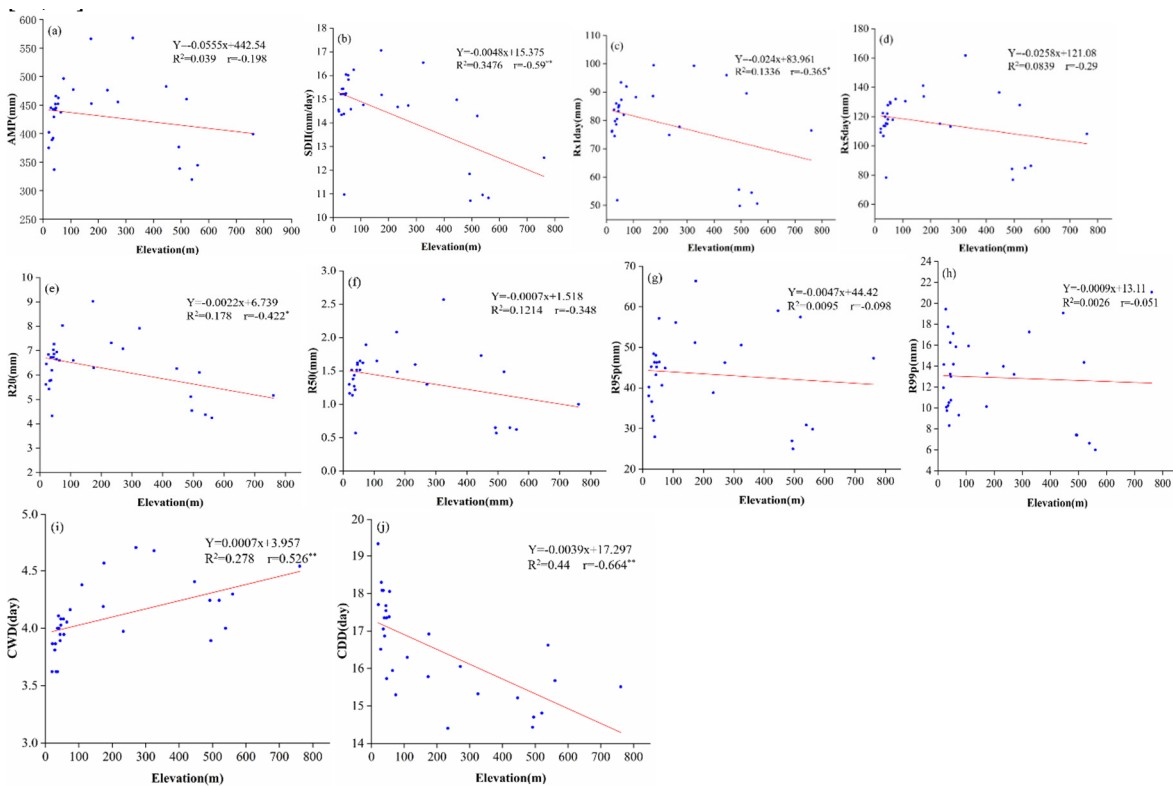

**Figure 10.** Relationship between elevation and precipitation indices for (**a**) annual mean precipitation (AMP), (**b**) simple daily intensity index (SDII), (**c**) number of moderate precipitation days (R20mm), (**d**) number of violent precipitation days (R50mm), (**e**) maximum 1-day precipitation amount (Rx1day), (**f**) maximum 5-day precipitation amount (Rx5day), (**g**) precipitation on very wet days (R95p), (**h**) precipitation on extremely wet days (R99p), (**i**) maximum consecutive dry days (CWD) and (**j**) maximum consecutive wet days (CDD).

## 5. Conclusions

Based on the daily and hourly precipitation data from 1981 to 2017, the comprehensive spatiotemporal variability of precipitation in Beijing was estimated and analyzed using multiple methods in this study. The main conclusions can be summarized as follows:

(1) The average annual precipitation in wet seasons showed a downward trend, while the SDII showed an upward trend during 1981–2017. Specifically, annual precipitation in the 2000s was significantly lower than in the past two decades, while it showed an upward trend during 2010–2017. Spatially, annual precipitation in the central urban area showed an obvious change during the study period. In the first part of the 21st century, the amount of annual precipitation in the central urban area was almost as great as that in Miyun county, which was the storm center during the previous three decades.

(2) Among the 10 precipitation indices, the precipitation indices which related to flood events, such as the SDII, Rx1day, R50mm, R95p, and R99p, showed a slight upward trend, which means the possibility of floods increased during 1981–2017; the average values of the annual maximum 3- and 6-h precipitation in the 2010s were higher than those in the past three decades, especially in urban and suburb areas compared to mountainous areas.

(3) The ENSO index has a considerably positive influence on the precipitation pattern, while the EASM index has a weakening effect on the pattern of wet seasons precipitation in Beijing. The precipitation pattern in the central urban area and surrounding suburbs in Beijing might be related to the change of impervious areas, and the topography was confirmed as one of the most important factors affecting the precipitation pattern in Beijing.

On the basis of the above, this research not only verified the conclusions of former researchers, but also made some new discoveries. The finding that the annual precipitation was decreased in the past few decades is consistent with findings from Zhai et al. and Song et al. [16,18], and the amount of precipitation in the central urban area is indeed slightly higher than in mountainous areas, and this trend is more obvious after the 2010s. However, the amount of extreme precipitation was increased during the study periods and this conclusion is different from the findings of previous studies; the extreme precipitation that occurred in 2012 and 2016 could be the main causes for this. There are many factors that affect regional precipitation distribution and variations, especially in urban areas. Both global climate change and regional water cycles affect the temporal and spatial variations of local precipitation. However, the physical mechanism of precipitation variation is highly complicated [28] and beyond the scope of this study. Instead, possible correlations between the atmospheric circulation index, local factors (urbanization and topography), and precipitation were discussed in this study. It is well known that several other factors (i.e., higher aerosol concentration, urban heat island effects, and large surface roughness) are important, and these will be further investigated in future studies.

**Author Contributions:** M.R. designed the technical routes of the study; M.R. analyzed the data and wrote the manuscript; Z.X., B.P., and J.L. proposed suggestions to improve the quality of the paper; and L.D. provided the observation data. All authors have read and agreed to the published version of the manuscript.

**Funding:** This work was financially supported by the National Key R&D Program of China (2017YFC1502701) and the National Natural Science Foundation of China (51879008). The authors wish to thank the Beijing Hydrology Bureau for providing the ground observation data.

**Conflicts of Interest:** The authors declare no conflicts of interest.

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
