# Peer review of "Spatiotemporal Variability of Precipitation in Beijing, China during the Wet Seasons"

_water, doi:10.3390/w12030716_

Round 1
Reviewer 1 Report
The paper is interesting and well-written. However, there can be found a few mistakes in the text that should be corrected.
The references are incorrectly cited in the text. For example, in line 34 the IPCC report should have the number [7] in the references instead of [6], and in line 39 – [8,9,10] instead of [7,8,9]. This incorrect numbering is present in the whole article. Please check and correct the references.
According to the references, please correct the 21st position – the authors’ first names are given instead of surnames.
Lines 55-58: „The second conclusion is that the phenomenon of a “precipitation island effect” was identified in Beijing, for example, Zhai et al. [16] and Zhen et al. [17] proposed that the average annual precipitation in central urban areas in Beijing is greater than that in surrounding areas because of urbanization effects.” In this sentence, the verb „proposed” seems to be inappropriate. Maybe the better verb would be „suggested”.
Lines 61-65: The sentence starting with „To comprehensive evaluate the changes in precipitation patterns(...)”. The fragment „(...) especially, Beijing has suffered the heavy rainstorms in 2012 and 2016(...)” is a parenthesis. It would be worth giving it in brackets – the sentence would be much clearer.
Line 72: In the heading, the word „description” seems to be unnecessary.
Line 89: „Location and spatial distribution of average annual precipitation in Beijing”. Figure 1 shows the location of observation stations and the spatial distribution of average annual precipitation. The title of this figure should, therefore, be completed by „Location of observation stations and spatial distribution (...)”.
In the subsection „Precipitation data” it would be good to state with what accuracy the data were recorded. I guess it was 0.01 mm?
In Table 2, the definitions of CWD and CDD should sound as follows: „Maximum number of consecutive days (...)” instead of „Maximum number of days consecutive days (...)”.
In the formulas (1) and (3) it should be an equal sign instead of a hyphen.
In the left part of Figure 2, the „2020” on the horizontal axis has been cut off. Please correct it.
In the right part of Figure 3, the values on the horizontal axis should not be in bold.
Figure 4 can be completed with the unit of annual average precipitation (mm).
The last chart in Figure 5 has got the description of the horizontal axis with a smaller font than in other charts.
Line 214: It seems to be a mistake in the number of the figure. It should be „Figure 6” instead of „Figure 7”. According to the same sentence, I suggest rewriting the beginning to: „It can be seen from Figure 6 that 40% of the stations (12 stations) presented an upward trend (...)”.
Line 229: The average Z value of the R95p seems to be not right. In Table 4, there is a value equal to -0.08 instead of -0.88. Which value is correct? Please check it.
Lines 244-245: Maybe it would be better to write „1-hour/3-hour/6-hour precipitation” instead of 1 hour, 3 hours and 6 hours precipitation?
Please consider a slight enlargement of axis descriptions in Figure 7, especially the vertical axis.
Line 314: I suggest to include the numbers given for drawings within Figure 9, for example: „Comparison of changes of land use in Beijing (a, c) and extreme precipitation indices in urban areas (b)”. It seems to me that the indication of figure (c) is written with a capital letter instead of a lowercase letter.
In lines 344 and 359 there are a few typos: „an slightly upward trend” instead of „a slightly upward trend”; „It is well know that there is a highly complicated mechanisms” instead of „It is well known that there are highly complicated mechanisms”. Please correct it.
Author Response
#1
Comments and suggestions for authors:
The paper is interesting and well-written. However, there can be found a few mistakes in the text that should be corrected.
The references are incorrectly cited in the text. For example, in line 34 the IPCC report should have the number [7] in the references instead of [6], and in line –[8, 9 , 10] instead of [7, 8, 9]. This incorrect numbering is present in the whole article. Please check and correct the references.
Response 1: Thank you for your kind comments. We have revised these mistakes in the manuscript.
According to the references. Please correct the 21st position – the authors’ first name are given instead of surnames.
Response 1: Thank you for your kind comments. We have revised this mistake in the manuscript.
“Pinskwar I.; Chorynski A.; Graczyk D.; Kundzewicz Z.W. Observed changes in extreme precipitation in Poland: 1991-2015 versus 1961-1990. Theor. Appl. Climatol. 2019, 135:773,787. http://doi.org/10.1007/s00704-018-2372-1.”
Lines 55-58: The second conclusion is that the phenomenon of a “precipitation island effect” was identified in Beijing, for example, Zhai et al. [16] and Zhen et al. [17] proposed that the average annual precipitation in central urban areas in Beijing is greater than that in surrounding areas because of urbanization effects”. In this sentence, the verb, “proposed” seems to be inappropriate. Maybe the better verb would be “suggested”.
Response: Thank you for your kind comments. We have changed this verb to “suggested” in the manuscript.
“For example, Zhai et al.[18] and Zhen et al.[19] suggested that the average annual precipitation in the central urban area in Beijing is greater than that in the surrounding areas because of urbanization effects.”
Line 61-65: The sentence starting with “To comprehensive evaluate the changes in precipitation patterns(…)”. The fragment(…) especially, Beijing has suffered the heavy rainstorms in 2012 and 2016(…)” is a parenthesis. It would be giving it in brackets- the sentence would be much clearer.
Response: Thank you for your kind comments. We have changed this sentence as follow.
“The objective of this study is to provide a better understanding of the characteristics of precipitation patterns and variations of extreme precipitation in Beijing (in particular, Beijing suffered heavy rainstorms in 2012 and 2016), and examine the influence of the atmospheric circulation index and local factors.”
Line 72: In the heading, the word “description” seems to be unnecessary.
Response: Thank you for your kind comments. We have deleted the word “description” from the heading in this line.
Line 89: Location and spatial distribution of average annual precipitation in Beijing. Figure 1 shows the location of observation stations and the spatial distribution of average annual precipitation. The title of this figure should, therefore, be completed by “Location of observation stations and spatial distribution (…)”.
Response: Thank you for your kind comments. We have changed the title of the Figure 1 to “Figure 1. Location of observation stations and spatial distribution of average annual precipitation in Beijing”.
In the subsection “Precipitation data” it would be good to state with what accuracy the data were recorded. I guess it was 0.01mm?
Response: Thank you for your kind comments. We have revised this content in the manuscript.
“The accuracy of both datasets was 0.01mm.”
In Table 2, the definitions of CWD and CDD should sound as follows: maximum number of consecutive days (…)” instead of “maximum number of days consecutive days (…)”.
Response: Thank you for your kind comments. We have modified the definitions of CWD and CDD in the manuscript.
In the formulas (1) and (3) it should be an equal sign instead of a hyphen.
Response: Thank you for your kind comments. We have revised formula (1) and formula (3).
In the left part of Figure 2, the 2020 on the horizontal axis has been cut off. Please correct it.
Response: Thank you for your kind comments. We have revised this mistake.
In the right part of Figure 3, the values on the horizontal axis should not be in bold.
Response: Thank you for your kind comments. We have revised this mistake.
Figure 4 can be completed with the unit of annual average precipitation (mm).
Response: Thank you for your kind comments. We have added the unit of annual average precipitation in the Figure 4.
The last chart in Figure 5 has got the description of the horizontal axis with a smaller font than in other charts.
Response: Thank you for your kind comments. We have modified this mistake.
Line 214: It seems to be a mistake in the number of the figure. It should be Figure 6 instead of Figure 7. According to the same sentence, I suggest rewriting the beginning to:, It can be seen from Figure 6 that 40% of the stations (12 stations) presented an upward trend (…)”
Response: Thank you for your kind comments. We have rewrite this sentence.
“It can be seen from Figure 6 that 40% of the stations (12 stations) presented an upward trend of the AMP indices (most of these stations were located in the central urban area and the southern suburb).”
Line 229: The average Z value of the R95P seems to be not right. In Table 4, there is a value equal to -0.08 instead of 0.88. Which value is correct? Please check it.
Response: Thank you for your kind comments. We have checked the average Z value of the R95p and changed this number to -0.08 in the Line 229.
Line 244-245: Maybe it would be better to write. 1-hour/3-hour/6-hour precipitation instead of 1 hour, 3 hours and 6 hours precipitation?
Response: Thank you for your kind comments. We have changed 1 hour, 3 hours and 6 hours precipitation by 1-hour/3-hour/6-hour precipitation in line 245.
Please consider a slight enlargement of axis descriptions in Figure 7, especially the vertical axis.
Response: Thank you for your kind comments. We have modified this Figure in the manuscript.
Line 314: I suggest to include the numbers given for drawings within Figure 9, for example: Comparison of changes of land use in Beijing (a, c) and extreme precipitation indices in urban areas (b). It seems to me that the indication of figure (c) is written with a capital letter instead of a lowercase letter.
Response: Thank you for your kind comments. We have modified the caption of Figure 9 and the relative contents in the manuscript.
“Figure 9. Comparison of changes of land use in Beijing (a,c) and extreme precipitation indices in urban areas (b). ”
“The temporal and spatial changes of land use during 1980–2017 in Beijing are shown in Figure 9–(a).”
In line 344 and 359 there are a few typos: “an slightly upward trend” instead of “a slightly upward trend”, It is well know that there is a highly complicated mechanisms instead of It is well known that there are highly complicated mechanisms. Please correct it.
Response: Thank you for your kind comments. We have revised these two errors.

Reviewer 2 Report
Dear Editor of the Water Journal.
I am very excited to have been given the opportunity to revise the manuscript entitled: “Spatiotemporal variability of precipitation in Beijing, China during the flood seasons”.
The manuscript by Ren and co-workers describes the author’s investigations into the spatial and temporal variability of precipitation in Beijing during the flood seasons.
The manuscript is generally good, but it needs to be revised, organized better and rewritten again.
The introduction and methods of work have been well formulated, but the results need to be reproduced, as well as better interpreted, especially the sections, 3.1, 3.2 and 3.3.
Major Comments:
The manuscript requires significant proof reading and revision to improve the quality of English. For example, the introduction has several long sentences which need to be completely restructured. There are also several grammatical mistakes in the manuscript. These types of errors occur throughout the manuscript and will need to be addressed. I think the manuscript needs significant editing for language and writing quality.The article lacks a comparison between the results obtained and the previous results in the same region especially about the relationship between the precipitation extremes and teleconnection patterns. Also, there is no information about the teleconnection patterns, topography and precipitation in the introduction
Dea
The article is based on pure statistical analyses without mentioning the causes of some of the results obtained (physical, geographic or climatic reasons, for example). I think it is better to explain it further.
Minor Comments:
L16. The SDII. I suggest to be written by the full name for the first time to be clear for the reader (Simple daily intensity index, SDII).
L20. Please replace the word “were” with the word “was”.
L30. I suggest deleting the word “US “.
L53. Please replace the word “shows” with the word “show” for present or could be written as “showed” for the past.
L 54 to L58. Very long sentence (4 lines), please Split it.
L 80. I suggest rewriting this sentence as it was mentioned in L92.
L95. Which precipitation data? It should be clear for reader.
L97 and L 98. Beijing was divided into six different areas according to the previous studies. Which criteria were used? The sentence is not clear.
L 106. Please replace the word “were” with the word “are”.
L 106. Please remove the “land use”.
L106 to L 110. It is a very long sentence. Please split it. I suggest rewriting it again.
L 115. I would like to add the word “extreme” to be precipitation extreme indices. Please.
L 118. Please add the word “extreme”.
L 118. Please replace the word “were” with the word “are”.
Table 2. I suggest adding the indicator name to all indices to be clearer.
L 131. Please replace the word “adopted” with the word “adopts”.
L 133 and L 134. I suggest removing these two lines from this paragraph.
L 146. Why the author used the Spearman's rank correlation coefficient?
L163. Flood seasons (June to September). This period is one season. I suggest to change the expression (Flood season) into wet season.
L168. Please rewrite this sentence.
L 270. Please rewrite it.
L 275 to L278. Please rewrite this paragraph again. The word negatively was repeated several times in two lines.
L 278 and L 279. Please rewrite it.
Tables and Figures:
Please provide more complete information in the caption of Figures and Tables. Captions are often too short and not complete. For example Figure 1 and 2 are not clear, if the left or the right is for the entire year.
Finally, I propose to reformulate the manuscript in accordance with the title and objectives of the article first before giving the decision in its publication.
Author Response
#2
The introduction and methods of work have been well formulated, but the results need to be reproduced, as well as better interpreted, especially the sections, 3.1, 3.2 and 3.3.
Major comments:
The manuscript requires significant proof reading and revision to improve the quality of English. For example, the introduction has several long sentences which need to be completely restructured. There are also several grammatical mistakes in the manuscript. These types of errors occur throughout the manuscript and will need to be addressed. I think the manuscript needs significant editing for language and writing quality.
The article lacks a comparison between the results obtained and the previous results in the same region especially about the relationship between the precipitation extremes and teleconnection patterns. Also, there is no information about the teleconnection patterns, topography and precipitation in the introduction.
Response: Thank you for your kind comments. We have added the comparison between the results obtained and the previous results about the relationship between the precipitation extremes and teleconnection patterns in the manuscript.
“Previous studies have shown that ENSO events have considerable impact on extreme precipitation in China, and the East Asian Summer Monsoon (EASM) also has an important moderating effect on precipitation in the eastern parts of the China [34-35]. For example, Fu et al. (2013) suggested that the annual precipitation and the frequencies of extreme precipitation are correlated with the ENSO index across China, while the relationship is different at different time scales and during different time periods [36]. Miao et al. (2019) concluded that the regional means of R95p were positively correlated with the ENSO across China during 1957–2014[37].”
“The results indicate that the ENSO has a considerable influence on the AMP, SDII, R95p, and R99p indices, while the EASM index had a weakening effect on the AMP, SDII, R95p, and R99p indices in the wet seasons during 1971–2017 in Beijing. These findings are similar to previous research results in Beijing; for example, Wei et al. (2017) analyzed the correlation between the extreme precipitation indices and large-scale climate variables in the Beijing–Tianjin sand source region. They found the ENSO and EASM index had a significant impact on precipitation extremes during 1960–2014[38]. Song et al. (2019) concluded that there was a significant relationship between the EASM and the extreme precipitation indices in Beijing during the end of the 1970s and in the 1980s [14].”
The article is based on pure statistical analyses without mentioning the causes of some of the results obtained (physical, geographic or climatic reasons, for example). I think it is better to explain it further.
Response: Thank you for your kind comments. We have added these contents in the manuscript.
“Both global climate change and regional water cycles affect the temporal and spatial variations of local precipitation. However, the physical mechanism of precipitation variation is highly complicated [28] and beyond the scope of this study. Instead, possible correlations between the atmospheric circulation index, local factors (urbanization and topography), and precipitation were discussed in this study. It is well known that several other factors (i.e., higher aerosol concentration, urban heat island effects, and large surface roughness) are important, and these will be further investigated in future studies.”
Minor Comments:
L16. The SDII. I suggest to be written by the full name for the first time to be clear for the reader (Simple daily intensity index, SDII).
Response: Thank you for your kind comments. We have modified this content in the manuscript.
“It was concluded that the average annual precipitation in wet seasons showed a downward trend, while the simple daily intensity index (SDII) showed an upward trend.”
L20 Please replace the word “were” with the word “was”.
Response: Thank you for your kind comments. We have revised this mistake.
L30 I suggest deleting the word “US”.
Response: Thank you for your kind comments. We have deleted the “US” from the manuscript.
L53 Please replace the word “shows” with the word “show” for present or could be written as “showed” for the past.
Response: Thank you for your kind comments. We have revised this mistake in the manuscript.
L54 to L58 Very long sentence (4 lines), please split it.
Response: Thank you for your kind comments. We have revised this sentence.
“The first is that both annual precipitation and extreme precipitation in Beijing show a downward trend in the past decades. For example, Song et al. [17] found that the frequency, amount, and contributions of extreme precipitation events in Beijing have had significant downward trends in the previous 50 years (1960-2012). The second conclusion is that the phenomenon of a “precipitation island effect” has occurred in Beijing. For example, Zhai et al. [18] and Zhen et al. [19] suggested that the average annual precipitation in the central urban area in Beijing is greater than that in the surrounding areas because of urbanization effects.”
L80. I suggest rewriting this sentence as it was mentioned in L92.
Response: Thank you for your kind comments. We have revised this sentence in the manuscript.
“The average annual precipitation in Beijing ranges from 372.1 to 682.9 mm/year at different observed stations, based on daily observed precipitation data at 30 stations from 1981 to 2017.”
L95. Which precipitation data? It should be clear for reader.
Response: Thank you for your kind comments. We have revised this content in the manuscript.
“Strict quality control was performed on both daily and hourly precipitation data”.
L97 and L 98. Beijing was divided into six different areas according to the previous studies. Which criteria were used? The sentences is not clear.
Response: Thank you for your kind comments. We have revised this content in the manuscript.
“In this study, to reflect the characteristics of precipitation in different subareas characterized by distinct topographic conditions, Beijing was divided into six different subareas.”
L106. Please replace the word “were” with the word “are”.
Response: Thank you for your kind comments. We have modified this mistake in the manuscript.
L106. Please removed the “land use”.
Response: Thank you for your kind comments. We have deleted the “land use” from this sentence.
L106 to L110. It is a very long sentence. Please split it. I suggest rewriting it again.
Response: Thank you for your kind comments. We have revised this sentence in the manuscript.
“These data were obtained from both the remote sensing monitoring database of land use status in China (Data Center for Resources and Environmental Sciences of the Chinese Academy of Sciences), and the Landsat remote sensing image database (www.gscloud.cn). The image data from 1980 were collected from the remote sensing monitoring database of land use status in China, which based on interpretations of MODIS and Landsat-TM satellite remote sensing images [30], the resolution of this image data is 1 km. The data of the remaining years were derived from Landsat series images, and the resolution of these images data are 30 m.”
L115. I would like to add the word “extreme” to be precipitation extreme indices. Please.
Response: Thank you for your kind comments. We have added the word “extreme” in this line.
L118. Please add the word “extreme”.
Response: Thank you for your kind comments.
“In this study, annual mean precipitation and nine extreme precipitation indices are adopted to analyze the spatiotemporal variation of extreme precipitation in Beijing.”
L118. Please replace the word “were” with the word “are”.
Response: Thank you for your kind comments.
“In this study, annual mean precipitation and nine extreme precipitation indices are adopted to analyze the spatiotemporal variation of extreme precipitation in Beijing.”
Table 2. I suggest adding the indicator name to all indices to be clearer.
Response: Thank you for your kind comments. We have added the name of indices in Table 2.
Table 2. Indices of extreme precipitation.
|
Code |
Descriptive name |
Definition of the indices |
Units |
|
AMP |
Annual mean precipitation |
Annual mean precipitation |
mm |
|
SDII |
Simply daily intensity index |
Annual mean precipitation/ total number of wet days |
mm/day |
|
R20mm |
Number of moderate precipitation days |
Annual precipitation days with daily precipitation greater than 20mm |
day |
|
R50mm |
Number of violent precipitation days |
Annual precipitation days with daily precipitation greater than 50mm |
day |
|
Rx1day |
Maximum 1-day precipitation amount |
Annual maximum 1-day precipitation |
mm |
|
Rx5day |
Maximum 5-day precipitation amount |
Annual maximum 5 consecutive days of precipitation |
mm |
|
R95p |
Precipitation on very wet days |
Annual precipitation exceed 95% threshold |
mm |
|
R99p |
Precipitation on extreme wet days |
Annual precipitation exceed 99% threshold |
mm |
|
CWD |
Maximum consecutive wet days |
Maximum number of consecutive days with daily precipitation greater than or equal to 1.0mm |
day |
|
CDD |
Maximum consecutive dry days |
Maximum number of consecutive days with daily precipitation less than 1.0mm |
day |
L131. Please replace the word “adopted” with the word ”adopts”.
Response: Thank you for your kind comments. We have modified this mistake in the manuscript.
“This study adopts several methods to analyze the spatiotemporal variation of extreme precipitation in Beijing.”
L133 and L 134. I suggest removing these two lines form this paragraph.
Response: Thank you for your kind comments. We have modified this sentence in the manuscript.
“The Kriging interpolation method was used to calculate the spatial distribution of annual precipitation in this study.”
L146 Why the author used the Spearman’s rank correlation coefficient?
The Spearman’s correlation coefficient can assess how well the relationship between two time series data. This method does not assume a linear dependence between data series, and thus, it allowed us to test for more general monotonic relationships. Therefore, we used this method in this study.
L163. Flood seasons (June to September). This period is one season. I suggest to change the expression (Flood season) into wet season.
Response: Thank you for your kind comments. We have changed the “flood seasons” to “wet seasons” in the manuscript.
L168. Please rewrite this sentence.
Response: Thank you for your kind comments. We have revised this sentence.
“The maximum annual precipitation in the 1980s and 1990s appeared in Miyun county, which is located in the northeast of Beijing. However, in the 2000s, the storm center gradually extended to the central urban area, and at the beginning of the 21st century the central area became the second storm center in Beijing, since its annual precipitation was almost as great as that in Miyun county.”
L270. Please rewrite this sentence.
Response: Thank you for your kind comments. We have revised this sentence.
“The ENSO index was positively correlated with AMP at all stations, and approximately 87% of stations had positive correlations between the ENSO index and SDII indices. The ENSO index exhibited statistically significant positive correlations with the R95p and R99p indices at 28 and 26 stations, respectively.”
L275 to L278. Please rewrite this paragraph again. The word negatively was repeated several times in two lines.
Response: Thank you for your kind comments. We have rewrite this sentence.
“Conversely, the AMP and SDII indices at most of stations were negatively correlated with the EASM index, and there were 29 and 28 stations that showed statistically negative correlations with the EASM index.”
L278 and L279. Please rewrite it.
Response: Thank you for your kind comments. We have rewrite this sentence.
“The results indicate that the ENSO has a considerable influence on the AMP, SDII, R95p, and R99p indices, while the EASM index had a weakening effect on the AMP, SDII, R95p, and R99p indices in the wet seasons during 1971–2017 in Beijing.”
Tables and Figures:
Please provide more complete information in the caption of Figures and Tables. Captions are often too short and not complete. For example Figure 1 and 2 are not clear, if the left or the right iis for the entire year.
Response: Thank you for your kind comments. We changed the name of Figure 1 to “Figure 1. Location of observation stations and spatial distribution of average annual precipitation in Beijing”.
We changed the name of Figure 2 to “Figure 2. Temporal variation of average annual precipitation across the whole year (a) and in the wet seasons (b)”.
Finally, I propose to reformulate the manuscript in accordance with the title and objectives of the article first before giving the decision in its publication.

Round 2
Reviewer 2 Report
No comment for the authors